# Impact of COVID-19 on the Anxiety Perceived by Healthcare Professionals: Differences between Primary Care and Hospital Care

**DOI:** 10.3390/ijerph18063277

**Published:** 2021-03-22

**Authors:** Ana C. Londoño-Ramírez, Sandro García-Pla, Purificación Bernabeu-Juan, Enrique Pérez-Martínez, Jesús Rodríguez-Marín, Carlos J. van-der Hofstadt-Román

**Affiliations:** 1Servicio de Farmacología Clínica, Hospital General Universitario de Alicante, Departamento de Farmacología, Pediatría y Química Orgánica, Universidad Miguel Hernández, C/Pintor Baeza 12, 03010 Alicante, Spain; anacarolina.londono@gmail.com; 2Instituto de Investigación Sanitaria y Biomédica de Alicante (ISABIAL), C/Pintor Baeza 12, 03010 Alicante, Spain; eperezmar@yahoo.es (E.P.-M.); rod.marin@umh.es (J.R.-M.); cjvander@umh.es (C.J.v.-d.H.-R.); 3Servicio de Salud Mental, Hospital Vega Baja, Carretera Orihuela-Almoradi s/n, 03314 San Bartolomé, Spain; sandrogarciapla@gmail.com; 4Unidad de Psicología Hospitalaria, Hospital General Universitario de Alicante, C/Pintor Baeza 12, 03010 Alicante, Spain; 5Servicio de Salud Mental, Hospital General Universitario de Alicante, Departamento de Medicina Clínica, Universidad Miguel Hernández, C/Pintor Baeza 12, 03010 Alicante, Spain; 6Departamento de Psicología de la Salud, Universidad Miguel Hernández, Avda de la Universidad s/n Edificio Altamira, 03202 Elche, Spain

**Keywords:** COVID-19, anxiety, health personnel, primary care, hospital care

## Abstract

The COVID-19 pandemic has had an emotional impact on healthcare professionals at different levels of care, and it is important to understand the levels of anxiety of hospital personnel (HP) compared to those of primary care personnel (PCP). The objectives herein were to assess the differences in anxiety levels between these populations and to detect factors that may influence them. The anxiety levels (measured using the Hospital Anxiety and Depression (HAD) scale) of the HP and PCP groups were compared using data collected from a cross-sectional study. The secondary variables included demographic and health data, confinement factors, contact with COVID-19 patients, having suffered from COVID-19, perceptions of protection, caregiver overload, threat, and satisfaction with management. We found anxiety “case” (35.6%) and “at-risk” (21%), with statistically significant differences in the group “at risk”, and higher scores in the PCP group. The factors associated with the perception of threat and protection were significant determinants of an increase in anxiety, with all of them showing statistically significant differences. There were greater symptoms of anxiety in the PCP group than the HP group (32% vs. 18%). The factors associated with the prevalence of anxiety symptoms were the perceptions of threat, protection, management, caregiver overload, and perceived degree of threat associated with COVID-19.

## 1. Introduction

The coronavirus disease 2019 (COVID-19) pandemic, which is affecting practically the entire world population at present, has affected the psychological health of the entire population in general, with an increase in emotional alterations of varying degrees of intensity and severity [1].

In Spain, from the start to the end of the State of Alarm (from 14 March to 21 June 2020), during which the present study began, the official data indicated that 244,109 individuals had become infected with the disease, and a total of 27,136 individuals had died from it; moreover, 124,642 had needed hospitalization, and a total of 11,620 had required an intensive care unit (ICU) stay [2]. As such, healthcare professionals had to deal with an extraordinary emotional load while working in the healthcare system, with very intense stressors such as the lengthening of their work days, work overload, a permanent state of alarm, fear of a possible infection, and the scarcity of protective equipment, at least during the start of the pandemic, as well as the obligation to perform tasks and activities for which they were not sufficiently prepared [1,3]. Some studies have revealed that, in the context of the COVID-19 pandemic, the work overload, as well as the fear of a possible infection, could have created and/or increased the levels of stress of healthcare professionals, with the subsequent appearance of anxiety [4]. In Wuhan, China, the prevalence of anxiety among the healthcare professionals on the front lines was found to be 38.5%, with a significantly higher prevalence in female healthcare workers on the front lines, those who received insufficient training and protection equipment, those who lacked confidence about the protection measures, and those worried about being infected [5].

Similar research studies on epidemics in the past have shown that healthcare professionals do not tend to suffer a psychological impact that is similar to the general population, but instead suffer a higher impact, showing signs of anxiety, depression, sleep disorders, fear of death, and feelings of sadness and irritability [6,7,8,9,10,11]. Similarly, a recent meta-analysis found similar results associated with the COVID-19 pandemic in healthcare professionals [12]. Studies performed in our country have revealed that, during the State of Alarm, 53% of healthcare workers showed values comparable to post-traumatic stress after the first wave of hospital care due to COVID-19 [13].

A research study on intensive care unit personnel indicated that working in an environment of continued contact with suffering and death, where important decisions have to be made with urgency, takes a psychological toll on healthcare personnel, with the level of depression reaching 20%, anxiety 7%, risk of suffering from fatigue due to compassion 12%, and burnout 3% [14].

In Spain, the health service structure is composed of hospital care and primary care centers. Hospital care personnel (HP) are centered on pathologies that are more diagnostically and therapeutically complex, while primary care personnel (PCP) are centered on the care of patients with chronic and prevalent pathologies.

Faced with sudden changes in patient care needs during the State of Alarm due to COVID-19, both groups have had to re-shape the way they work, designing strategies and processes to provide pertinent and necessary health services, and have had to incorporate these into their habitual care services. In this sense, the PCP group is centered on the detection and monitoring of cases, while the HP group is centered on the treatment of more severe cases [1,3].

Given that studies related to the emotional alterations and state of anxiety of healthcare professionals have not considered the possible differences between the two different work teams (HP and PCP), after experiencing a state of health emergency due to COVID-19, our present work intended to assess the presence or risk of anxiety in these two groups.

Due to the different activities performed by the HP and PCP groups, we considered it of great importance to assess the psychological impact between them after the end of the first period of the State of Alarm due to COVID-19, to plan guidelines and interventions necessary for maintaining the psychological well-being of these professionals. Thus, the principal objective of this study was to determine if there are differences in the anxiety levels between these groups. In the case of differences, as a secondary objective, we aimed to identify if these differences could be determined based on the risk of COVID-19 infection, the perception of threat, and the measures of protection, and to detect factors of vulnerability that could have an influence on the presence of anxiety in these healthcare personnel.

## 2. Materials and Methods

### 2.1. Design

We performed a cross-sectional, descriptive, and observational study to compare the HP and PCP groups (including medicine, psychology, nursing, social work, laboratory technicians, radiology, nurse assistants, administrators, caretakers, maintenance, and cleaning personnel) who had cared for COVID-19 patients, by exploring variables that could be associated with the prevalence of symptoms of anxiety in this population.

### 2.2. Participants

Healthcare professionals were invited to participate (HP and PCP) through the “communications network from the Alicante Health Department-General Hospital” (email, intranet, and social networks). Participation was voluntary, anonymous, and without compensation. The recruitment procedure was the same in both groups, and at the beginning of the self-administered questionnaire, each professional indicated the group in which he/she belonged. The inclusion criteria were working in the department during the COVID-19 crisis, voluntary acceptance, and signing a voluntary consent form to participate; the exclusion criterion was answering fewer than 80% of the items in the questionnaire.

A sample size of 70 participants per group was calculated (*n* = 140), estimating a Cohen’s d effect size of 0.33 (mean difference), a level of significance of 0.05, and a power of 0.8. The initial number of participants was 351 individuals, although eight records were posteriorly excluded (2.3%) by meeting the exclusion criteria, leaving a final sample of 343 individuals.

### 2.3. Procedure

The data were collected through a Google Form between 26 June and 6 July 2020. The first page of the form sought to collect the participants’ consent. Data were collected anonymously. The link was distributed amongst the professionals at the department through the normal information channels—intranet of the department, emails from the Specialized Care and Primary Attention offices, and social networks—to attain the greatest number of professionals. Additionally, for data collection, the requirements set by the Organic Law 3/2018, from 5 December, for the protection of personal Data, the guarantee of digital rights and the general European Union regulation of data protection 2016/679, were followed. The study obtained the approval of the ethics committee from the General University Hospital of Alicante (code: PI2020-104).

### 2.4. Measurements

The following data were obtained: sociodemographic data (age, gender, marital status, level of education, and professional category), clinical history and health habits (smoking, medical history, alcohol consumption, and use of anxiolytics or anti-depressants), factors associated with confinement (living alone or with family, with individuals older than 70 years old or younger than 18, type of housing, having quarantined, or changes in the usual way of working, conducted in person or online), contact with COVID-19 patients at work, having been infected with the disease, perceptions of protection, caregiver overload, threat, and satisfaction with the management of health services. To assess their perceptions, an ad-hoc questionnaire was designed, whose main questions for the subjects are shown in Table 1. For the assessment of perceptions, for all items, affirmative responses were scored 1, with a greater score indicating a greater degree of perception.

Additionally, two open-ended questions were asked. The first was introduced in the perception of management item, which assessed if the work had been performed as usual, with a dichotomous response (yes/no). In the case of a negative answer, they were asked what had changed, to discover the main changes in the level of work associated with COVID-19. The second question inquired about aspects that generated worry: “What has worried you the most, or what is worrying you about the current situation?”

To measure the level of anxiety, we utilized the Hospital Anxiety and Depression (HAD) scale [15] in the Spanish version by Terol et al. [16]. The cutoff points established were “no case” (<8; no anxiety symptoms), “at-risk” (8–10; at risk of developing anxiety), and “case” (>10; developed anxiety).

### 2.5. Data Analysis

We analyzed the data with SPSS version 25 (SPSS Inc., Chicago, IL, USA). The description of the data was performed through absolute (n) and relative frequencies for categorical variables; for the continuous variables with a normal distribution, the mean and standard deviations were utilized, while the median was used for the quantitative variables with a non-normal distribution. To compare the categorical and continuous objectives between the two groups, Pearson’s *χ*^2^ and Student’s *t*-tests were utilized, respectively. For the analysis of the influence of work conditions and risk of infection, multiple logistic regression was utilized.

## 3. Results

### 3.1. Sample

Of the 343 participants, 265 belonged to the HP group (77.3%) and 78 to the PCP group (22.7%). The average age was 47.16 years old (SD = 11.95), with an age range between 22 and 69 years; 264 were women (76.9%). No differences were observed according to the professional categories or other sociodemographic factors between the PCP and HP groups (Table 2). Of the HP, 150 worked in medical specialty services, 78 worked in emergencies, 16 in the ICU, and 21 in other services (radiology and laboratory).

### 3.2. State of Health

We did not find differences between the PCP and HP groups with respect to their medical histories, use of anxiolytics or anti-depressant drugs (Table 3), and the toxic habits evaluated.

### 3.3. Factors Associated with Anxiety

From the total sample, 35.6% of the participants were considered “cases” of anxiety, and 21% were considered to be “at risk” of anxiety. The percentage of “cases” in the PCP and HP groups was similar—37% and 35%, respectively—with statistically significant differences in anxiety symptoms (“at-risk”) between PCP (32%) and HP (18%) (Pearson’s *χ*^2^ test = 0.008) (Figure 1).

After analyzing the factors that could be associated with anxiety symptoms (“at-risk”) or anxiety disorders (“case”), significant associations were not observed for the variables age, marital status, and level of education. However, the female group showed greater levels of anxiety (“at-risk” and “case”). Significant differences were also found between the professional categories, with a low effect size. The least-affected group was medical staff, while the most affected was nursing staff and technicians (Table 4).

As for their state of health, only 3.5% (*n* = 12) of our sample had previous psychopathological antecedents, and all of these subjects showed anxiety symptoms (“at-risk”). Significant differences were not found in the levels of anxiety between the groups with or without medical antecedents (Table 3).

Significant differences were not observed in the levels of anxiety influenced by the presence of COVID-19 infection. Moreover, differences were not observed associated with having been in quarantine, the fact that changes were made to the habitual manner of working, or the fact that the activity was conducted in person or online.

Lastly, differences were not observed in the state of anxiety associated with the factors of confinement (confinement with the family, as a couple, with friends, or alone; persons younger than 18 or older than 70 at home; type of housing where one was confined).

From the total sample, 206 (68.8%) subjects had direct contact with infected individuals. Of these, only 8% were in contact with family members, while the rest had contact with patients (34.9%), colleagues (30.1%), or both (31.1%). We found significant differences in the levels of “at-risk” and “case” anxiety associated with being in contact with infected individuals (*p* = 0.013). From this group, 22.2% (*n* = 46) were scored as “at-risk” and 40.6% (*n* = 84) as “case”.

The factors associated with perceptions (Table 2) were the most determinant of an increase in anxiety (“at-risk” and “case”). All of these showed statistically significant differences (most with a large effect size) in the perception of threat, perception of protection, perception of management, perception of caregiver overload, perception of threat perceived associated with COVID-19, and satisfaction with the management of the center (Table 5).

When evaluating the aspects that created worry, we found that the main worries in our sample were infection (34.6%), uncertainty and lack of information (13.7%), lack of equipment (12.8%), and the resurgence of the disease (12.2%).

### 3.4. Factors Related to the Prevalence of Anxiety Symptoms (“at-Risk” and “Case”)

After the univariate analysis of anxiety symptoms with the variables of “gender,” “professional category,” and “perceptions of protection, management, overload, anxiety related to the protection measures, threat, and satisfaction with the protection at the center,” with statistically significant results (previously described in Table 4 and Table 5), we realized a multiple logistic regression, controlled by “gender” and “professional category.” We observed a greater risk of anxiety symptoms (“at-risk” and “case”) of 2.4 times in women, and we also observed statistically significant results in the variables “perceptions of threat” (3.7 times) and ”perception of anxiety related to protection measures” (5.5 times) (Table 6).

## 4. Discussion

In this study, we compared two homogeneous populations of healthcare workers who had provided care to the population during the first wave of the COVID-19 pandemic while performing their duties at hospitals and primary care centers (HP and PCP, respectively), and we also assessed their levels of anxiety according to the HAD scale. The findings indicated a greater anxiety symptomology (“at-risk”) in the PCP group. When analyzing the different variables implicated in anxiety, we observed that the factors associated with its prevalence were the perception of threat, perception of protection, perception of management, perception of caregiver overload, and perceived degree of threat associated with COVID-19.

The degree of anxiety in the general sample in our study (56.5%) was similar to that found in Spanish healthcare workers during the COVID pandemic (58.6%) [17], although lower than that found in a study in Barcelona, where 71.6% showed anxiety symptoms [18], and higher than another study conducted in the autonomous communities of the Basque Country and Navarra, where 37% showed anxiety symptoms [2]. In both cases, the differences could be related, as the data collection was conducted during the first months of the pandemic (March/April and April, respectively), while in our case, it was conducted at the end of the State of Alarm in June. Moreover, in these other two cases, the instrument utilized to anxiety evaluation was the DASS-21, which could have had an influence on these differences. Another possible explanation for these differences could be the occurrence of the disease in the centers where the data were collected.

In our study, we found a greater level of anxiety in the PCP compared to the HP. These differences do not depend on the pre-existence of greater psychopathology, since the results were similar in both groups, comparing antecedents in psychopathology and the use of anxiolytic and antidepressant medications. Perhaps if we consider that primary care is the door to the health system, uncertainty and caregiver overload could have been greater in this group, and this could explain our results. In the literature reviewed, we found that anxiety among healthcare workers can be attributed to a long work day, a lack of personal protection equipment, and fear of infecting oneself or one’s family [19,20,21]. In our results, we also found an influence of these factors on the level of anxiety in the population studied, which could have been extraordinary in the PCP. Nevertheless, we could not ascertain if differences in these factors between the HP and PCP existed, as an intragroup multivariate analysis could not be performed, given the size of the sample.

In our results, the variables “female” and “nurse” showed a greater level of anxiety, just as other in studies conducted in healthcare worker populations [2,17,18,22,23,24]. In this sense, we should be cautious when making conclusions in this respect, as 77% of the sample was composed of women; however, these results coincide with a previous prevalence study on anxiety and depression in the Spanish population, where women obtained greater percentages than men in all anxiety-related disorders (7.6% as compared to 2.6%) [25].

An important aspect of our study was to determine the influence of the perception of threat and the satisfaction with the protection received by the health personnel on the development of anxiety. Many studies have concluded that the perception of risk related to disease is associated with worse mental health results, especially in the current pandemic, as well as previous resurgences of other pandemics [26,27,28,29].

According to our results, the perception of threat and the satisfaction with the degree of protection did not vary between the PCP and HP groups, although variations were found in the factors that had an influence on the levels of anxiety. These data are coherent with the results from some of the studies reviewed. In a meta-analysis of 62 studies from 17 countries, including Spain, Luo et al. [30] found that the worries due to the threat of infection and the scarcity of protection equipment contributed to a greater level of anxiety.

Additionally, we found a significant relationship between the levels of anxiety of our subjects and clinical antecedents, especially the psychopathological antecedents. This also coincides with the reviewed literature [23,31,32]. Given that all of the participants who had psychological antecedents had high levels of anxiety, it is important to improve the treatment and monitoring of this highly vulnerable population.

## 5. Conclusions

In our study, a greater level of anxiety was found in the PCP than in the HP, and thus we believe it is relevant to consider this variable in future studies directed toward healthcare worker populations. In this sense, we highlighted that there is a lack of studies that evaluate the impact of COVID-19 on the mental health of healthcare professionals that differentiate both groups, and we also pointed out that the existing studies were conducted at a critical point during the pandemic, while the present study was conducted posteriorly.

We are in agreement with Erquicia et al. [18] on the need for intervention plans at hospitals for the healthcare professionals who require them, and in our case, we believe that special attention should be given to the PCP. The data of this study could be taken into consideration for the planning of programs related to the prevention and psychological care of healthcare personnel, as higher-risk groups were identified.

## 6. Limitations

The limitations of this study are centered on the sample, as we utilized a population from a single health department, so the results cannot be generalized. Likewise, the data collection procedure utilized could have been used only by individuals who had greater anxiety levels or were more motivated by the subject, and this aspect could not be controlled, so future studies are needed to eliminate these biases.

The lack of data around years of work experience and/or resilience as a covariate is a limitation to illustrating the impact of the anxiety levels reported in this study. Given the anonymous management of data, we could not guarantee that a participant would not complete the questionnaire multiple times. Another limitation was that the ad-hoc questionnaire was not tested for face/content validity and test–retest reliability, so there could be misclassification bias.

## Figures and Tables

**Figure 1 ijerph-18-03277-f001:**
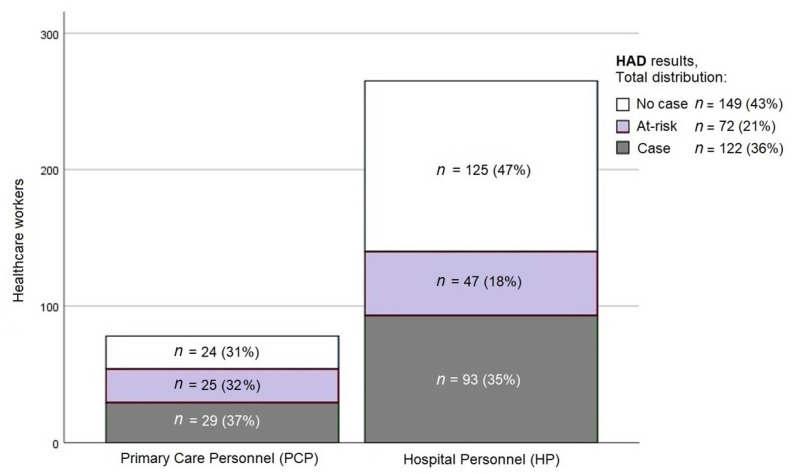
Levels of anxiety between the PCP and HP. HAD, Hospital Anxiety and Depression scale.

**Table 1 ijerph-18-03277-t001:** Questions that assess the perceptions of threat, protection, caregiver overload, perceived threat, and satisfaction with management associated with coronavirus disease 2019 (COVID-19).

**Perception of Protection**	Have you felt sufficiently protected to face the situation?
**Perception of management**	Do you believe the situation has been well managed at your health service?
	Have you performed your work in the habitual manner?
**Perception of overload**	Have your perceived a work overload in this situation?
**Perception of anxiety related to the protection measures**	Do you feel anxious due to the matters of disinfection and protection?
**Perception of threat**	Have you ever thought about quitting due to the appearance of COVID-19?
	Have you ever thought that your family and friends have avoided contact with you due to your work?
	Have you been worried that the members of your family could be infected by COVID-19?
	Have you been worried that you could be infected by COVID-19?
**Satisfaction with the protection at the center**	Are you satisfied with the protection measures against nosocomial infection provided by your service?
	Are you satisfied with your work schedule?
	Are you satisfied with the logistics support provided by your hospital or health center?

**Table 2 ijerph-18-03277-t002:** Sociodemographic data of the sample.

		PCP (*n* = 78)	HP (*n* = 265)	*p* *
Age	Mean years old (SD)	51 (±11)	46 (±12)	
Gender	Male	22 (28.2%)	57 (21.5%)	0.410
	Female	56 (71.8%)	208 (78.4%)	
Marital status	Married	47 (60.3%)	141 (53.2%)	0.454
	Single	17 (21.8%)	85 (32.1%)	
	Widowed	2 (2.6%)	4 (1.5%)	
	Separated	12 (15.3%)	34 (12.8%)	
Education	Primary school	0 (0%)	7 (2.6%)	0.105
	High school	11 (14.1%)	57 (21.5%)	
	University	67 (85.9%)	201 (75.8%)	
Professional category	Doctor	38 (48.7%)	82 (30.9%)	0.103
	Nurse	22 (28.2%)	83 (30.9%)	
	Nursing assistant and other technicians	6 (7.7%)	42 (15.8%)	
	Caretaker	3 (3.8%)	9 (3.4%)	
	Administrative staff	6 (7.7%)	30 (11.3%)	
	Other	3 (3.8%)	17 (6.4%)	

* Pearson’s *χ*^2^ test (significance, *p* < 0.005). PCP, primary care personnel; HP, hospital personnel.

**Table 3 ijerph-18-03277-t003:** Health state characteristics of the sample.

		PCP (*n* = 78)	HP (*n* = 265)	*p* *
Antecedents	AH and cardiovascular	9 (11.5%)	29 (10.9%)	0.707
	Respiratory	4 (5.1%)	12 (4.7%)	
	Psychopathological	4 (5.1%)	9 (3.4%)	
	Various diseases	11 (14.1%)	28 (10.6%)	
	Other	12 (15.4%)	31 (11.7%)	
	No pathology	38 (48.7%)	156 (58.9%)	
Drugs				
	Use of anxiolytics	23 (29.5%)	51 (19.2%)	0.094
	Use of antidepressants	4 (5.1%)	19 (7.2%)	0.542
	Both anxiolytics and antidepressants	2 (2.56%)	7 (2.64%)	0.225

* Pearson’s *χ*^2^ test (significance, *p* < 0.005). PCP, primary care personnel; HP, hospital personnel; AH, arterial hypertension.

**Table 4 ijerph-18-03277-t004:** Sociodemographic characteristics and state of anxiety.

	Anxiety	No Case (*n* = 149)	At-Risk (*n* = 72)	Case (*n* = 122)	*p* *	Intensity of Association (Cramer’s V)
Gender	Male	54 (67.5%)	15 (18.8%)	11 (13.8%)	<0.001	0.286
	Female	95 (36.1%)	57 (21.7%)	111 (42.2%)		
Professional	Doctor	68 (56.7%)	25 (20.8%)	27 (22.5%)	0.003	0.172
category	Nurse	35 (33.3%)	23 (21.9%)	47 (44.8%)		
	Technicians and assistants	15 (31.3%)	9 (18.8%)	24 (50.0%)		
	Caretaker, administrative staff, and others	31 (44.3%)	15 (21.4%)	24 (34.3%)		

* Pearson’s *χ*^2^ test (significance, *p* < 0.005). Doctors: physicians and psychologists.

**Table 5 ijerph-18-03277-t005:** Relationship between the level of anxiety and the perceptions of protection, management, caregiver overload, degree of threat perceived associated with COVID-19, and satisfaction with protection at the center.

	Anxiety	No Case (*n* = 149)	At-Risk (*n* = 72)	Case (*n* = 122)	*p* *	Intensity of Association (Cramer’s V)
Perception of protection	Protection	86 (65.2%)	23 (17.4%)	23 (17.4%)	<0.001	0.360
	Abandonment	63 (29.9%)	49 (23.2%)	99 (46.9%)		
Perception of management	Adequate	113 (53.8%)	45 (21.4%)	52 (24.8%)	<0.001	0.302
	Inadequate	36 (27.1%)	27 (20.3%)	70 (52.6%)		
Perception of overload	Yes	68 (34.5%)	43 (21.8%)	86 (43.7%)	<0.001	0.224
	No	81 (55.5%)	29 (19.9%)	33 (24.7%)		
Perception of anxiety related to the protection measures	Anxiety	34 (19.0%)	40 (22.3%)	105 (58.7%)	<0.001	0.561
	No anxiety	115 (70.1%)	32 (19.5%)	17 (10.4%)		
Perception of threat	Perceive threat	50 (25.1%)	49 (24.6%)	100 (50.3%)	<0.001	0.446
	No threat	99 (68.8%)	23 (16.0%)	22 (15.3%)		
Satisfaction with the protection at the center	Satisfied with protection	122 (48.6%)	54 (21.5%)	75 (29.9%)	<0.001	0.205
	Dissatisfied with protection	27 (29.3%)	18 (19.6%)	47 (51.1%)		

* Pearson’s *χ*^2^ test (significance, *p* < 0.005).

**Table 6 ijerph-18-03277-t006:** Multiple logistic regression. Analysis of anxiety symptoms (“at-risk”) according to factors of gender, professional category, and perception.

Related Factors	Odds Ratio	(95% CI)	*p*
Gender (female/male)	2.4	1.2–4.7	0.010
Professional category:			
Caretaker, administrative staff, and others			
Doctor	0.7	0.3–1.4	1.4
Nurse	1.0	0.5–2.2	2.2
Technicians and assistants	0.8	0.3–2.1	2.1
Perception of abandonment	0.6	0.3–1.2	0.161
Perception of inadequate management	0.9	0.5–1.6	0.619
Perception of overload	1.7	1.0–3.0	0.068
Perception of anxiety related to the protection measures	5.5	3.1–9.9	<0.001
Perception of threat	3.7	2.1–6.5	<0.001
Perception of dissatisfaction with protection	0.9	0.5–1.8	0.826

## Data Availability

The data presented in this study are available on request from the corresponding author. The data are not publicly available due there are unpublished results.

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
