# Peer review of "Impact of COVID-19 on the Anxiety Perceived by Healthcare Professionals: Differences between Primary Care and Hospital Care"

_ijerph, 2021, doi:10.3390/ijerph18063277_

Round 1
Reviewer 1 Report
The abstract and introduction (lines 71-76) suggest that this study will try and identify vulnerability and risk factors for Covid related anxiety in HP and PCP groups respectively. It did not do this as it mainly analysed the two groups combined rather than separately. Thus the aim of identifying if anxiety was determined by the risk of covid, by the perception of threat and by protective measures as well as identifying vulnerability facors was not achieved.
It is unclear how he PCP group was recruited. For those living outside of Spain it is important to explain this aspect of the medical service.
I am unclear as to what is meant by the "perception of anxiety related to protection measures" (number 4 of the 6 perception measures) as distinct from "anxiety related to protection measures". Surely the two are similar.
Table 3 is unclear and appears to repeat "antidepressants prescribed" in lines 1 and 3 of that section. Also should the section heading not be "psychotropic medication" rather than antidepressants.
In lines 195-197 the authors write that significant differences were not observed in resepct of being under quarentine etc. These are not metinoned in the methods section and no data is presented on these variables.
Lines 213-215 are confusing. The first part regarding the individuals who percieved caregiver overload of whom 21.8% were at risk and 43.7% were cases of anxiety appears to be describing a specific line in table 5. The next sentence about caregiver load among those with anxiety refers I think to a column with an n=122. Is this correct? Why did the authors focus on this rather than doing the same for each of the percepion questions? If there is a specific reason it should be stated.
In lines 216-18 the authors mention aspects that create "worry" - what is their definition of worry (question5)? Why was the lesser symptom of worry introduced? They mention variables such as lack of equipment - does this equate with "protection measures" mentioned in question 6 of the questionnaire. Resurgence of disease is new information and as far as I can see is not mentioned elsewhere in the paper.
In table 6 they mention prevalence. Surely they mean odds ratio?
In table 6 only the statistically significant variables in the multivariae analysis are really relevant and these are being female, "perception of anxiety related to protection" and "perception of threat". This should be stated clearly.
The authors rightly say that caution must be exercised in respect of the high prevaence of anxiety in women in this study. An interaction between gender and profession would have assisted but due to inadequate power this type of detailed analysis could not be conducted.
I do not see any guidelines in this discussion other than the general one of monitoring those who had prior anxiety disorder (lines 71-73). Preparation of guidelines was one of the goals. Data on prior anxiety and it's role in covid-related anxiety was not presented anywhere except in table one comparing the PCP and HP groups.
Author Response
Please, see the attachment

Reviewer 2 Report
Thank you for the opportunity to review this interesting manuscript. Overall the details were presented coherently to answer the aim of the proposed study. I have a few comments for the authors to consider:
- Abstract:
- Methods:
- Line 21: Please include the object that was compared, i.e. “Anxiety levels (measured using HAD) of the HP and PCP groups were compared using data collected from a cross-sectional study.”
- The phrase “and descriptive” (line 21); “and level of anxiety (HAD)” (line 24); can be deleted.
- Results:
- Please rephrase the first sentence (line 24-26) – it is not clear what “risk” is. Please revise to “at risk”. Since it is translated from Spanish, the authors may quote the HAD score instead, e.g. “at risk” (HAD 8-10) to be consistent with the methods used.
- Please do not begin a sentence with number.
- The second sentence (line 26-27) may be rephrased to “Factors associated with the perception of threat and protection were significant determinant of increase in anxiety (adjusted OR = xxx, 95% CI of the adjusted OR = xxx, xxx).”.
- The authors stated that “There were greater symptoms of anxiety in the PCP group” (line 28). Please state the actual results, e.g. how much more? E.g. 9/10 vs 5/10?
- Materials and Methods:
- Section 2.2:
- Please indicate if the effect size of 0.33 (line 93) was based on mean difference or risk ratio.
- Line 94: Please correct the power - 0.08 is very low, it is likely to be 0.8.
- Lines 95-101 (along with Table 1) belongs in the Results section. Please remove from here.
- Section 2.3: Please indicate how the participants provided consent. Was it included in the first page of the Google Form? Also, were the participants given their specific de-identified ID to complete the form? If not, how would the research team ensure that a participant did not complete the questionnaire multiple times thus swaying the average score to theirs and/or giving a false representation of the ‘actual’ responses?
- Section 2.4:
- Please clarify if the ad-hoc questionnaire was tested for face/content validity and test-retest reliability. If not, please acknowledge this as a limitation of the study in the Discussion section. There could be misclassification bias.
- It would be better to include both the HAD score and description when referring to the categories (line 130). For instance, “The cutoff points established were “no case” (<8; no anxiety symptoms), “at risk” (8-10; at risk of developing anxiety) and “case” (>10; developed anxiety).” It would be good to update the terminology in the legend of Figure 1 and throughout the manuscript as well.
- Section 2.5:
- Please replace “qualitative variables” with “categorical variables” and replace “quantitative variables” with “continuous variables” (lines 157-158).
- Please revise “multivariate regressions” to “multiple logistic regression” (line 161).
- Results:
- Table 1: Please replace “x” with “mean” or use the correct symbol with a bar above the x.
- Table 4: The authors could report the Chi-squared statistics instead of Cramer’s V.
- Tables 4 and 5: Please revise p-value of “0.000” to “<0.001” because p-value is never zero.
- Please only keep 1 decimal place when reporting % in tables. Please also update the text throughout the manuscript.
- Tables 5 & 6: Please run a single multiple logistic regression with all the factors, including “gender” and “professional category” in the model because they were significant from the univariate analyses. Running multiple comparisons increases the chance of significance when there may be none. Please also update the text in the results section accordingly.
- Discussion:
- It would be interesting to compare with the other literature in regards to the (high) use of drugs amongst the study participants.
- Please update the discussion accordingly after revising the results from the multiple logistic regression instead of referring to the results of the multiple regression models.
- In addition to the limitation noted above, please include another one, i.e. lack of data around years of work experience and/or resilient as a covariate to illustrate the impact of anxiety level reported in this study.
- Section 2.2:
- Methods:
Author Response
Please, see the attachment

Reviewer 3 Report
In this work, authors present a cross-sectional and descriptive study aimed to investigate the differences in anxiety levels between hospital and primary care personnel and to detect factors that may influence them. I find this paper interesting due to provide relevant information according the situation we are living right now. However, some suggestions are proposed to improve the work.
Abstract:
Results section is a little bit confusing and conclusions section is repetitive. Please, consider to rewrite it in a clearer way.
Introduction:
This section contains all of the information necessary for readers to understand the purpose of the study. Nevertheless, some references are missing from some statements you provide (page 1, lines 34-36 and page 2, lines 63-66)
Please, provide – in the “COVID 19”. What ICU means?
Material and methods:
Why 8 records were posteriorly excluded?
I am never convinced by providing results in the participant’s section (page 4, line 96). It would help the reader if you report it at result section.
I do not understand the sentence “Between the HP, 78 worked in emergencies, 16 in the ICU/REA, and 21 in other services” (line 98). What about the remaining 150?
In table 1, what does “x” refer to?
I find the ad hoc questionnaire description (table 2) a little bit confusing. It is not very clear how all the items have been scored.
What are the psychometric properties of the HAD?
Results:
In table 3, what HTN means?
The tables and figures are easy to interpret and understand. Nevertheless, I think that the text is not organized in a logical way, and it is difficult to follow the flow of ideas through the text. Please, consider to restructure it.
Data in figure 1 is different to the text (page 8, line 176).
Discussion:
Discussion is clearly and well written. You interpret the study’s findings in light of those of other published (and updated!) studies.
Conclusions only are focused on the first objective of the study. What about the factors associated and related to anxiety?
What are the strengths of this study?
Reviewer 4 Report
Topics researched by other researchers. However, today it is a very important scientific topic and it is worth conducting further research related to it also in the group of students of medical universities.
Round 2
Reviewer 1 Report
The authors have responded to my queries adequately